# In Vitro Characterization and Safety Assessment of *Streptococcus salivarius*, *Levilactobacillus brevis* and *Pediococcus pentosaceus* Isolated from the Small Intestine of Broiler Breeders

**DOI:** 10.3390/microorganisms13061231

**Published:** 2025-05-27

**Authors:** Nwabisa Happiness Kokwe, Freedom Tshabuse, Feroz Mahomed Swalaha

**Affiliations:** 1Department of Biotechnology and Food Science, Durban University of Technology, Durban 4001, South Africa; happykokwe12@gmail.com; 2Department of Biochemistry and Microbiology, University of Zululand, KwaDlangezwa 3886, South Africa

**Keywords:** gastrointestinal system, probiotics, lactic acid bacteria, poultry

## Abstract

In poultry production, antibiotics have been excessively used as growth promoters to support well-being and decrease mortality caused by pathogenic microorganisms. The overuse of antibiotics has led to the emergence of antibiotic-resistant bacteria and the presence of antibiotic residues in poultry products. To counteract this problem, probiotics could be used as adjuncts or as substitutes for preserving a diverse and balanced microflora to prevent the colonization and multiplication of pathogenic bacteria in the GI tract. This study aimed to isolate and characterize the potential probiotic properties of lactic acid bacteria from the small intestine of 23-week-old broiler breeders, with the goal of identifying potential probiotic candidates. Four phenotypically healthy broiler breeders were selected, and intestinal contents were aseptically collected and cultured on MRS agar. From the initial pool of 39 colonies, six isolates were identified based on Gram-positive and catalase-negative characteristics and further classified using 16S rRNA sequencing as *Levilactobacillus brevis* (*n* = 3), *Pediococcus pentosaceus* (*n* = 2), and *Streptococcus salivarius* (*n* = 1). These strains were further evaluated for probiotic properties such as transit resistance to simulated upper gastrointestinal conditions, antagonist activity, haemolytic activity, and cell surface properties such as autoaggregation, co-aggregation and hydrophobicity, *in vitro*. *L. brevis* NKFS8 showed good tolerance to pH 3, while *P. pentosaceus* NKSF10 exhibited good tolerance to pH 4 acidic conditions. All isolates demonstrated good survivability in bile salt concentration of 3% (*w*/*v*), with *P. pentosaceus* NKSF10 exhibiting the highest tolerance. The isolates showed a wide range of antagonistic activity against the test pathogens *Pseudomonas aeruginosa* (ATCC 27853), *Salmonella typhimurium*, *Salmonella enterica* (ATCC 13314), *Staphylococcus aureus* (ATCC 29213), and *Listeria monocytogenes* (ATCC 7644). Furthermore, these strains exhibited good auto-aggregation, co-aggregation, and hydrophobicity properties. In conclusion, lactic acid bacteria from the small intestine of broiler breeders present a valuable prospect for the development of effective probiotics. These probiotics can be utilized as a supplementary inclusion in poultry feed, obviating the need for antibiotics as growth promoters. Nevertheless, additional in vivo studies are required to closely monitor and assess the effects of probiotics on the gastrointestinal system of chickens.

## 1. Introduction

The poultry sector plays an essential role in South Africa as it provides affordable protein for millions of households and enhances the South African agricultural economy [1]. As reported by the South African Poultry Association (SAPA) [2], the poultry industry accounts for 19.6% of total agricultural gross value and 40% of animal products, making it the largest contributor to the agricultural sector. However, over the years, the industry has been facing critical challenges such as the cost of feed, the scale of production and disease outbreaks [2]. To counteract these challenges, antibiotics have been extensively used in poultry production to enhance productivity, improve growth performance, and control infections. However, this practice has also led to the emergence of antibiotic residues in poultry products and the spread of antibiotic-resistant bacteria in meat [3]. Moreover, the poultry industry is rapidly growing, and various aspects such as breeding, nutrition, and animal health require attention to improve production efficiency. In intensive poultry production systems, the health and proper functioning of the avian gastrointestinal tract (GI) are crucial for optimal poultry performance [4]. As a result, there has been an increase in the search for new alternatives, and the utilization of native microorganisms with probiotic capacity has the potential to offer an efficient alternative for the prevention of certain animal diseases [5,6].

Probiotics are defined as live microorganisms that confer health benefits to the host when administered in adequate amounts [7]. Among the most studied probiotics are lactic acid bacteria (LAB), which have been widely utilized in livestock farming as natural growth promoters, as they are generally recognized as safe (GRAS) [8]. LAB strains from the genera of *Lactobacillus*, *Enterococcus*, *Streptococcus*, *Pediococcus*, and *Bifidobacteria* have shown potential for improving gut health and reducing the need for the usage of antibiotics as growth promoters [9].

Probiotics support host health through multiple mechanisms, including maintaining gut microbial balance, regulating animal growth [10], and inhibiting pathogenic bacteria by producing antimicrobial substances such as organic acids, hydrogen peroxide, bacteriocins, and bacteriocin-like compounds [11]. Studies have highlighted their antagonistic effects against pathogens such as *Escherichia coli*, *Salmonella enteritidis*, *Staphylococcus aureus*, *Salmonella typhimurium*, and *Campylobacter jejuni* [12,13,14]. Probiotics further confer benefits through competitive exclusion, immune system stimulation, cell surface interactions, and the detoxification of harmful compounds via bacterial cell wall binding [15].

For a bacterial strain to qualify as a probiotic, it must meet stringent criteria, including survival under pH and bile salt conditions, antimicrobial activity against pathogens, adhesion to intestinal epithelial cells, and resilience during processing and storage [16]. Host-specific probiotics, particularly those isolated from natural habitats, such as the GI tract, are preferred because of their enhanced adaptability and efficacy within the host environment [9]. This underscores the need to identify and evaluate novel, host-specific strains to maximize health benefits and improve livestock productivity [9].

In this study, six LAB strains, *Levilactobacillus brevis* (*n* = 3), *Pediococcus pentosaceus* (*n* = 2), *and Streptococcus salivarius* (*n* = 1) were isolated from the small intestine of broiler breeders for detailed molecular and functional analysis. These strains were selected for their potential probiotic properties, including their ability to survive gastrointestinal conditions, inhibit pathogenic bacteria, and exhibit adhesion to the intestinal epithelium. The molecular characterization of these strains provides insight into their genetic and functional traits, which underpin their probiotic potential. Functional analyses, such as their antimicrobial activity and tolerance to bile salt and low pH, were performed to evaluate their applicability in improving poultry health and reducing dependence on antibiotics. This study bridges the gap between probiotic characterization and practical applications in the poultry industry.

## 2. Materials and Methods

### 2.1. Sample Collection

Small intestinal samples were obtained from four locally bred broiler breeders at a commercial slaughterhouse in the KwaZulu-Natal province, Ulundi, South Africa. Upon collection, samples were aseptically stored in sterile conditions and transported on ice to the laboratory for immediate processing. Sterile phosphate-buffered saline (PBS, pH 7.2) was used for initial sample handling to ensure microbial integrity and minimize contamination.

### 2.2. Isolation of Lactic Acid Bacteria (LAB) from the Gastrointestinal Tract of Broiler Chickens

LAB isolation was performed using a previously established method by [9] with slight modifications. Briefly, collected small intestine samples were rinsed with sterile PBS (pH 7.2) to remove residual debris. Ten grams of each sample (*n* = 4) were homogenized and subjected to enrichment in de Man Rogosa Sharpe (MRS) broth (Merck, Modderfontein, South Africa) under anaerobic conditions at 37 °C for 24 h with continuous shaking. Enriched samples were serially diluted in sterile 0.9% saline and plated on MRS agar (Merck). Anaerobic conditions were maintained using anaerobic jars containing Anaerogen™ sachets (Thermo Fisher Scientific, Johannesburg, South Africa) at 37 °C for 72 h. Distinct, white colonies were selected and purified through three successive subcultures on MRS agar. Preliminary identification of LAB was conducted by Gram staining and catalase testing, following the guidelines provided in Bergey’s Manual of Systematic Bacteriology [17]. Colonies showing Gram-positive and catalase-negative results were preserved in MRS broth supplemented with 28% (*v*/*v*) glycerol and stored at −80 °C for further analysis.

### 2.3. Molecular Identification of LAB by 16S rDNA Sequencing

The identification of LAB isolates was conducted through 16S rRNA gene sequencing. Briefly, total genomic DNA was extracted using the Quick-DNA™ Bacterial Miniprep Kit (Zymo Research, Catalogue No. D6005, Pretoria, South Africa), following the manufacturer’s instructions. The 16S rRNA gene was amplified via polymerase chain reaction (PCR) in 20 μL reaction mixture containing 10 μL of GoTaq 2× Green Master Mix, 0.5 μL each of universal primers 27F (5′ AGAGTTTGATCMTGGCTCAG 3′) and 1492R (5′ CGGTTACCTTGTTACGACTT 3′) at a concentration of 10 μM, 6 μL of nuclease-free water, and 2 μL of DNA template. PCR amplification was performed under the following thermal cycling conditions: initial denaturation at 94 °C for 5 min, followed by 35 cycles of denaturation at 94 °C for 30 s, annealing at 50 °C for 30 s, and extension at 68 °C for 1 min. A final extension step occurred at 68 °C for 10 min. Amplicons were visualized on a 1% (*w*/*v*) agarose gel in Tris-acetate-EDTA buffer, run at 80 V for 50 min. A 100 bp DNA ladder was used as a molecular weight marker. Amplicons of the desired size were purified and sequenced by Inqaba Biotech (Pretoria, South Africa). Sequence data were analyzed using BLASTn (version 2.14.0) on the NCBI platform (https://blast.ncbi.nlm.nih.gov/Blast.cgi) (accessed on 12 January 2024) with a minimum threshold of 99% query coverage and 99% identity. Phylogenetic trees were constructed using the neighbour-joining method implemented in MEGA 12 software to confirm species identity.

### 2.4. Preliminary Screening of LAB

#### 2.4.1. Assessment of Acid Tolerance in LAB Isolates

The acid tolerance of the LAB isolates was assessed following the protocol by [18]. Overnight LAB cultures (10^8^ CFU/mL) grown in MRS broth at 37 °C were inoculated into MRS broth adjusted to pH 3 and pH 4 using 1 N HCL. Cultures were incubated anaerobically at 37 °C for 3 h. Post-incubation, 100 µL aliquots were plated on MRS agar and incubated at 37 °C for 48 h. Survivability percentages were calculated using the equation:$$pH\;survivability\;\%=\frac{Viable\;LAB\;count\;\left(\frac{CFU}{mL}\right)after\;acid\;exposure}{Initial\;viable\;LAB\;count\;\left(\frac{CFU}{mL}\right)}$$

#### 2.4.2. Assessment of Bile Salt Tolerance in LAB Isolates

The bile salt tolerance of the LAB was determined following the modified method of [9]. LAB cultures (10^8^ CFU/mL) were inoculated into MRS broth supplemented with 0.3% (*w*/*v*) and 0.6% (*w*/*v*) bile salt (Merck) and incubated anaerobically for 3 h at 37 °C. Survivability was determined as described for acid tolerance.$$Bile\;salt\;survivability\;\%=\frac{Viable\;LAB\;count\left(\frac{CFU}{mL}\right)after\;bile\;salt\;exposure}{Initial\;viable\;LAB\;count\;\left(\frac{CFU}{mL}\right)}$$

### 2.5. Safety Assessment of LAB

#### 2.5.1. Assessment of Haemolysis Activity in LAB Isolates

LAB isolates were streaked onto blood agar plates containing 5% (*v*/*v*) defibrinated sheep blood and incubated at 37 °C for 48 h. Hemolysis was assessed by observing clear (*β*-hemolysis), greenish (*α*-hemolysis), or no changes (*Y*-hemolysis) around colonies, as explained in a study by [6]. Non-hemolytic isolates (*Y*-hemolysis) were selected for further analyses.

#### 2.5.2. Determination of Antibiotic Resistance Profiles

Antibiotic susceptibility was evaluated using the agar disc diffusion method [19]. LAB suspensions adjusted to McFarland’s standard 0.5 (10^8^ CFU/mL) were evenly spread on MRS agar plates. Antibiotics were selected based on local treatment protocols, and availability of CLSI interpretive criteria, including penicillin G (2 units), ceftriaxone (30 μg), ampicillin (25 μg), vancomycin (30 μg), oxacillin (1 μg), streptomycin (10 μg), chloramphenicol (30 μg), gentamicin (10 μg), erythromycin (10 μg), tetracycline (10 μg), novobiocin (30 μg), and ciprofloxacin (10 μg), were then placed on the surface of the plates. Plates were incubated anaerobically at 37 °C for 24 h, and inhibition zones were measured with a digital calliper. Results were interpreted based on the Clinical & Laboratory Standards Institute (CLSI) M100, 32nd Edition, 2022 guidelines.

### 2.6. Functional Characterization of LAB

#### 2.6.1. Antagonistic Activity

The antagonistic properties of LAB isolates were evaluated using an agar well diffusion assay. Overnight LAB cultures were centrifuged 10,000× *g* for 10 min, and 100 μL of cell-free supernatants were added to wells on Mueller Hinton (MH) agar (Merck) seeded with target pathogens (*Pseudomonas aeruginosa* (ATCC 27853), *Salmonella typhimurium*, *Salmonella enterica* (ATCC 13314), *Staphylococcus aureus* (ATCC 29213), and *Listeria monocytogenes* (ATCC 7644)). Inhibition zones were measured after anaerobic incubation at 37 °C for 24 h.

#### 2.6.2. Assessment of Extracellular Enzymatic Activity in LAB Isolates

Amylase and protease activities were assessed as per the study of [20]. For amylase activity, LAB isolates were cultured in MRS broth containing 0.25% starch and anaerobically incubated at 37 °C for 24 h. Subsequently, 30 µL of the overnight cultures were inoculated onto a paper disc, which was then placed on a nutrient agar plate and anaerobically incubated for 48 h at 37 °C. Zones of hydrolysis were measured to confirm the enzymatic activity. For protease activity, LAB isolates were sub-cultured in MRS broth and incubated anaerobically at 37 °C for 24 h. After the incubation period, 30 µL of the overnight culture was placed on a paper disc, which was then transferred onto nutrient agar plates supplemented with 1% skim milk. The plates were anaerobically incubated at 37 °C for 48 h, and the diameter of the halo surrounding each paper disc was measured using a digital calliper.

#### 2.6.3. Auto-Aggregation and Co-Aggregation

To evaluate the auto-aggregation capacity of LAB, overnight cultures of LAB isolates grown in MRS broth were centrifuged at 5000× *g* for 15 min. Following the centrifugation, the pellet cells were washed three times with PBS (pH 7.2) and subsequently resuspended in 5 mL of PBS. The suspension was then adjusted to an optical density of 0.500 at 600 nm (OD600). The bacterial suspension was incubated at 37 °C for 4, 8, 16, and 24 h, as described by the study of [21]. At each time point, the optical density of the upper phase was measured at 600 nm to assess bacterial sedimentation. The auto-aggregation coefficient (AC) was calculated using the following formula:$$AC(t)\;\%=\left[1-\left(\frac{{OD}_{f}}{{OD}_{i}}\right)\right]\times 100$$
where *OD*_initial_ is the initial optical density of the microbial suspension at 600 nm. *OD*_final_ is the optical density of the microbial suspension after 4, 8, 16 and 24 h of incubation.

The co-aggregation assay was conducted by centrifuging overnight cultures of LAB isolates grown in MRS broth at 6000× *g* for 15 min. Following centrifugation, the resulting pellet cells were washed twice with sterile PBS (pH 7.2) and adjusted to a concentration of 10^8^ CFU ml/L in the same buffer. A 2 mL aliquot of each LAB isolate were then mixed with an equal volume of test pathogen cultures, including *Pseudomonas aeruginosa* (ATCC 27853), *Salmonella typhimurium*, *Salmonella enterica* (ATCC 13314), *Staphylococcus aureus* (ATCC 29213), and *Listeria monocytogenes* (ATCC 7644)). The mixtures were incubated for 4 h at 37 °C. Control tubes comprising 4 mL of either the LAB isolates or pathogen culture alone were used for comparison. The optical density (*OD*_600_) of the mixture (*OD_mix_*) was measured and compared with the absorbance of the control tubes containing the probiotic strains (*OD_strain_*) following a 4 h incubation, following the protocol described by the study of [22]. The percentage of co-aggregation was determined using the formula:$$Co-aggregation\;\%=[1-{OD}_{mix\;}/\left({OD}_{strain}+{OD}_{pathogen\;}\right)/2]\times 100$$

#### 2.6.4. Cell Surface Hydrophobicity

The evaluation of cell hydrophobicity, aimed at determining the adherence of LAB isolates to hydrocarbons, was performed according to the method previously outlined by [23]. Briefly, LAB isolates were centrifuged at 4000× *g* for 30 min, and the resulting pellet was resuspended in 3 mL of PBS (pH 7.2) to adjust to concentration of 10^8^ CFU/mL. To assess hydrophobicity, 1 mL of n-hexadecane was added to the bacterial suspension. The mixture was vortexed for 1 min to ensure thorough interaction and then allowed to stand undisturbed for 15 min, leading to a two-phase separation. The absorbance of the aqueous phase was measured at 600 nm using a spectrophotometer to determine the bacterial affinity to the hydrocarbon. The following equation was used to calculate the bacterial affinity to the solvent (BATS):$$BATS\;\%=[1-\left(\frac{A_{30min}}{A_{0min}}\right)]\;\times \;100$$
where *A*_30_ is the absorbance after 30 min of incubation. *A*_0_ is the initial absorbance before incubation.

### 2.7. Statistical Analysis

All experiments were conducted in triplicate to ensure reproducibility and reliability of the data. Statistical analysis was carried out using a One-Way Analysis of Variance (ANOVA) to determine significant differences among treatment groups, with a significance threshold set at *p* < 0.05. To further evaluate specific differences between groups, particularly in comparison to the positive control used in this study, Dunnett’s multiple comparisons post hoc test was applied.

## 3. Results and Discussion

### 3.1. Isolation of Bacterial Strains and Identification of Lab Isolates by 16s rDNA Gene Sequencing

LAB constitute the most extensively studied group of probiotics because of their significant benefits, including immune modulation, pathogen inhibition, and maintenance of gut homeostasis [24]. While the therapeutic potential of probiotics is well documented, the strain-specific nature of probiotic properties necessitates the continued exploration of new strains [25]. In alignment with this objective, the present study focused on isolating and characterizing LAB strains from the small intestine of 23-week-old broiler breeders (*n* = 4). Using MRS agar, 39 colonies were initially isolated, and only 6 LAB isolates exhibited probiotic properties as they were identified as Gram-positive and catalase-negative, possessing rod-and-cocci morphologies (Table 1). According to the study of [26], the morphology and biochemistry of Gram-positive bacteria enable them to survive and thrive in the digestive tract of the host. This is due to the special element, teichoic acid, which is responsible for maintaining ion transport, cell wall integrity, and others contributing to the resistance of the bacteria to autolysis and maintaining external permeability. This is in accordance with the findings obtained by the study of [26], who isolated Gram-positive and catalase-negative LAB strains from the small intestines of ducks and indicated that LAB are generally characterized by their catalase-negative characteristics due to the absence of gas bubbles containing oxygen when in contact with hydrogen peroxide (H_2_O_2_).

Isolates exhibiting distinct morphological traits were selected and identified using 16S rRNA gene sequencing. The results demonstrated that the isolates were classified into three species: *Streptococcus* spp., *Levilactobacillus* spp., and *Pediococcus* spp. Specific isolates exhibited high sequence homology with database reference strains, such as 100% of isolate SI4 to *Streptococcus salivarius* (NR042776.1) and 99.85% similarity of isolates SI6, SI8, and SI9 to *Levilactoacillus brevis* (NR116238.1), while isolates SI23 and SI38 displayed 99,58% and 99,51% similarity to *Pediococcus pentosaceus* (NR042058.1), respectively (Table 1). The phylogenetic tree-based 16S rRNA sequences (Figure 1) of the selected six LAB isolates have been uploaded to NCBI, with the following accession numbers listed in Table 1. The phylogenetic tree shown in Figure 1 is the relationship between the isolated LAB strains (indicated by the black dots) and five strains obtained from the Genbank based on 16S rRNA sequence gene analysis. These findings align with previous reports of the isolation of *Pediococcus* spp. [27], *Levilactobacillus* spp. [28] and *Streptococcus* spp. [29] from various sources. Furthermore, these findings suggest that LAB are indigenous members of the gut microbiota of poultry species [30]. The isolates were further evaluated for probiotic potential, assessing their tolerance to acidic and bile salt conditions, pathogen inhibition, enzymatic activity, and adhesion characteristics.

### 3.2. In Vitro Characterization of LAB

#### 3.2.1. Acid Tolerance

The capacity of isolates to exhibit substantial tolerance to highly acidic conditions within a GI tract is an essential attribute for probiotic candidates [31]. This study evaluated the potential of the isolated LAB strains to survive in MRS broth with acidic pH adjusted to 3 and 4 after 3 h of exposure. As shown in Figure 2, all LAB strains showed varying resistance to different acidic conditions, and the survival percentages ranged from 57.5 ± 7.50% to 78.00 ± 2.00% at pH 3 and 57.00 ± 3.00% to 83.25 ± 1.81% at pH 4. *L. brevis* NKFS8 and *S. salivarius* NKFS6 exhibited the highest tolerance at pH 3, with a survival percentage of 78.00 ± 2.00% and 73.61 ± 4.62%, respectively. Similarly, *P. pentosaceus* NKFS10 and *P. pentosaceus* NKFS11, *L. brevis* NKFS7 and *L. brevis* NKFS9 exhibited the highest acid tolerance at pH 4 with a survival percentage of 83.25 ± 1.81%, 80.09 ± 7.95%, 74.50 ± 4.50%, and 74.00 ± 2.00%, respectively. Our results are consistent with the findings of [32] who reported an impressive survivability rate of *P. pentosaceus* SC28 and *L. brevis* KU15151 to be more than 90%. Supporting this, a proteomic analysis by [33] revealed that acid stress in *L. casei* triggers the induction of stress-response proteins and molecular chaperones, including GroEL, GroES, DnaK, Clp GrpE, hrA, and SGP, which collectively enhance bacterial resilience under acidic conditions.

Interestingly, it was noted that the survivability percentages of *Pediococcus* spp. at pH 4 was much higher than that of the positive control, *L. casei* ATCC 393, which had a survivability percentage of 76.00 ± 4.00%. The significant variation (*p* < 0.05) in the survivability percentages of the LAB is attributed to species- or strain-specific acid tolerance mechanisms, as explained by the study of [21]. These mechanisms encompass central metabolic pathways, proton pumps, modifications in cell membrane composition and cell density, as well as processes for DNA and protein damage repair and acid neutralization [34].

#### 3.2.2. Bile Salt Tolerance

LAB isolates were evaluated for their ability to survive bile salt concentrations as it is, according to [6], a prerequisite for successful colonization and metabolic activity in the gut of the host, assisting probiotic bacteria to effectively contribute to the intestinal microflora. The bile salt survivability of the isolated LAB strains was assessed at concentrations of 0.3% (*w*/*v*) and 0.6% (*w*/*v*), with 0.3% (*w*/*v*) representing the recommended threshold for probiotic screening [30].

A significant difference (*p* < 0.05) in the variation in the survivability percentages of the LAB isolates was observed, with the survivability percentage ranging from 53.76 ± 0.52% to 81.95 ± 5.85% at both 0.3% and 51.50 ± 1.50% to 69.61 ± 4.59% at 0.6% bile salt concentrations (Figure 3) [35]. *P. pentosaceus* NKFS10 and *P. pentosaceus* NKFS11 exhibited maximum survival percentages of 81.95 ± 5.85% and 76.92 ± 15.39% at bile salt concentrations of 0.3% (*w*/*v*), respectively. Comparable findings were documented by the authors of [36], who observed that *Pediococcus* spp. strains exhibited growth rates ranging from 82% to 100% at 0.3% bile salt concentrations. This tolerance is attributed to the bile salt hydrolase, an enzyme that mitigates the toxic effects of bile salt, thereby enhancing probiotic survival within the intestinal environment. It was also observed that there was a reduction in the survivability percentage of the LAB strains with increasing bile salt concentrations, with *L. brevis* NKFS7 demonstrating the lowest survivability of 51.50 ± 1.50% at 0.6% bile salt. A similar trend was reported by the authors of [37], who noted a reduction in bile resistance of the LAB isolate N32 from 58.61% at 0.3% bile salt to 52.11% at 0.5%, representing a decrease of 3.5%.

In summary, isolates showing a survivability rate of 50% and above, including *P. pentosaceus NKFS10*, *P. pentosaceus NKFS11*, *L. brevis NKFS9*, *L. brevis NKFS8*, *L. brevis NKFS7*, and *S. salivarius NKFS6*, were selected for further in vitro screening as they meet the standard thresholds for probiotic potential.

### 3.3. Safety Assessment

#### 3.3.1. Haemolysis Activity

The haemolytic activity of LAB was also determined as it is recommended by the FAO to assess the pathogenic nature of potential probiotic strains. All isolates were non-haemolytic (Table 2), meeting safety standards for probiotic candidacy as per FAO guidelines. A study by [38] also reported that all LAB were found to be gamma haemolytic. Haemolytic activity is a process of destroying red blood cells, leading to the release of soluble components and transparent red liquid [39]. Non-haemolytic behaviour supports their non-virulent nature, consistent with earlier findings in LAB studies [9,40]. *Lactobacillus casei* (ATCC 393) was used as the positive control for the selection of all non-haemolytic isolates.

#### 3.3.2. Antibiotic Susceptibility Testing

The global rise in antimicrobial resistance presents a critical and growing threat to human, animal, and environmental health. In this study, the antibiotic susceptibility profiles of LAB isolated from the GI of broiler breeders were assessed against a range of antibiotics (Table 2). All isolates exhibited resistance to ciprofloxacin, vancomycin, and novobiocin. Similarly, oxacillin resistance was universal across isolates, except for *P. pentosaceus* NKFS11, which exhibited intermediate resistance (I). Interestingly, penicillin susceptibility varied; *P. pentosaceus* NKFS10 and NKFS11 displayed resistance, while all other isolates exhibited intermediate, underscoring the diversity in beta-lactam susceptibility within LAB. Gentamicin resistance was noted in all isolates except *S. salivarius* NKFS6, which showed intermediate susceptibility. Streptomycin susceptibility varied significantly, with *L. brevis* NKFS8 being susceptible (S), *S. salivarius* NKFS6 and L. brevis NKFS9 showing intermediate susceptibility, and all other isolates exhibiting resistance. Antibiotic resistance observed among diverse LAB strains is often attributed to chromosomally encoded resistance determinants, aligning with previous reports that associate intrinsic resistance with the limited cellular uptake of antibiotics or modifications of target sites [41,42]. The current findings corroborate those of [43,44], who documented resistance in LAB to streptomycin, and [45], who reported vancomycin resistance in *Lactobacillus* species. Notably, resistance to vancomycin in *Lactobacillus* species has been linked to the synthesis of altered peptidoglycan precursors, whereby the terminal D-alanine-D-alanine dipeptide is replaced with a depsipeptide form, D-alanine-D-lactate, thus diminishing antibiotic binding affinity [6]. While such mechanisms may underlie the resistance patterns observed in the present study, it is important to note that resistance levels were not quantitatively assessed via minimum inhibitory concentration (MIC) assays. Moreover, in the absence of molecular analyses to detect resistance genes, the distinction between intrinsic and acquired resistance on these isolates remains unresolved. It was further observed that isolates exhibited universal susceptibility to erythromycin, ampicillin, and tetracycline, except for *P. pentosaceus* NKFS10, which showed resistance to all three antibiotics. Chloramphenicol, a broad-spectrum antibiotic, was effective against most isolates except *P. pentosaceus* NKFS10. Sensitivity to chloramphenicol was reported in LAB isolated from poultry and pigs [46], which exhibited resistance, indicating a potential difference in efflux pump activity or enzyme-mediated inactivation. These findings disagree with [47], who reported that, among the inhibitors of protein synthesis, tetracycline resistance has been reported to be the most common acquired resistance in food isolates of *Lactobacillus* and *Streptococcus*:

Gentamicin (GM), streptomycin (S), chloramphenicol (C), ceftriaxone (CRO), ciprofloxacin (CIP), vancomycin (VA), erythromycin (E), ampicillin (AMP), penicillin (P), oxacillin (OX), tetracycline (TE), novobiocin (NV). Resistant (R), intermediate (I), susceptible (S).

### 3.4. Functional Characteristics

#### 3.4.1. Antagonistic Activity of LAB

The ability of potential probiotic bacteria to inhibit pathogenic bacteria is an important criterion for the selection [48]. LAB isolates were tested for their ability to inhibit the growth of common poultry pathogens by agar well diffusion assay. All LAB strains in this study exhibited varying inhibitory effects against the Gram-positive and Gram-negative indicator pathogens, *L. monocytogenes*, *P. aeruginosa*, *S. typhimurium*, *S. enterica*, and *S. aureus*. Figure 4 represents a heatmap illustrating the antagonistic activity of the LAB isolates against selected pathogens, with the colour gradient ranging from purple (indicating weak inhibition) to yellow (indicating strong inhibition). The results demonstrate that *L. brevis* NKFS9, *L. brevis* NKFS7, and *S. salivarius* NKFS6 exhibited the most potent antagonistic effects against S. enterica, as indicated by the orange-yellow hues. The corresponding zones of inhibition were 30.00 ± 0.00 mm, 29.00 ± 1.41 mm, 28.50 ± 12.02 mm, and 28.50 ± 3.54 mm, respectively (Appendix A). These findings underscore the potential of these strains as effective antagonists against enteric pathogens.

Interestingly, these isolates were further able to show a slightly strong inhibition against *P. aeruginosa* with zones of inhibition of 24.50 ± 0.71 (*L. brevis* NKFS9), 23.50 ± 2.12 (*S. salivarius* NKFS6), 21.50 ± 0.71 (*L. brevis* NKFS7), and 21.00 ± 0.00 (*L. brevis* NKFS8). Another slightly strong inhibition was exhibited by *L. brevis* NKFS9, *P. pentosaceus* NKFS10, and *L. brevis* NKFS7 against *L. monoctogenes*, *S. typhimurium*, and *S. aureus* with a zone of inhibition of 25.00 ± 7.07, 23.00 ± 7.07, and 23.00 ± 4.24, respectively. These findings are in accordance with the study by [47], which reported a broad-spectrum inhibition of LAB isolated from donkey feces against pathogens. According to [49], this inhibitory effect is a result of the production of antimicrobials such as H_2_O_2_, lactic acid, bacteriocins, diacetyl, and a combination of various uncharacterized antimicrobial substances released into the cellular milieu of the pathogenic bacteria. Although this study employed the agar well diffusion assay to evaluate the antagonistic activity of LAB against pathogenic strains, it did not incorporate the molecular characterization of antimicrobial metabolites, nor did it control for confounding factors such as pH or organic acid production. As a result, it remains inconclusive whether the observed inhibitory effects can be attributed specifically to bacteriocin synthesis.

Minimum inhibition (represented by purple/blue hues) was observed in isolates *P. pentosaceus* NKFS11, *P. pentosaceus* NKFS10, and *S. salivarius* NKFS6 with the inhibitory zones of 13.50 ± 0.71 mm, 13.50 ± 0.71 mm, and 14.50 ± 0.71 mm against *S. enterica* and *L. monocytogenes*, respectively. Isolates *P. pentosaceus* NKFS11 and *P. pentosaceus* NKFS10 further displayed minimum inhibition against *S. aureus* and *P. aeruginosa* with zones of inhibition of 14.50 ± 0.71mm and 13.50 ± 0.71 mm. It was noted in this study that the antagonistic effect of the isolated LAB strains showed no relationship with the Gram classification of the pathogens. Our findings align with the findings reported in several studies [21,47,50,51].

#### 3.4.2. Extracellular Enzymatic Activity

Protease and amylase enzymes produced by LAB play a significant role in the digestion and absorption of nutrients in the GI tract [52]. LAB isolates were tested for their ability to produce dietary enzymes amylase and protease. As presented in Figure 5, the results were obtained by measuring the size of the halo zone surrounding the colonies as an indicator for extracellular activity, and all six isolates were able to produce both amylase and protease enzymes with varying zones of clearance ranging from 8.50 ± 4.95 mm to 10.50 ± 0.71 mm for protease and 9.50 ± 0.71 mm to 12.00 ± 1.41 mm for amylase. In contrast to our findings, Ref. [27] reported that *L. brevis* isolates from chickens were only protease-positive but extracellular-amylase-negative. *L. brevis* NKFS7 and *L. brevis* NKFS9 in this study exhibited the highest protease activity with zones of clearance of 10.50 ± 0.71 and 10.00 ± 1.41 mm, respectively. Interestingly, these isolates also exhibited the highest protease activity, with zones of clearance of 12.00 ± 1.41 mm (*L. brevis* NKFS9) and 11.00 ± 1.41 mm (*L. brevis* NKFS7). The production of enzymes by potential probiotic strains for broiler chickens can degrade antinutritional compounds present in poultry feed, subsequently improving body weight gain and feed conversion efficiency [53]. Additionally, amylase is essential for starch hydrolysis as it makes up a significant portion of chicken diets, while protease protein is essential for hydrolysing protein, which is advantageous in chicken diets consisting of amino acids and complex protein [54].

#### 3.4.3. Auto-Aggregation and Co-Aggregation

The evaluation of auto-aggregation and co-aggregation abilities of potential probiotic bacteria is essential as they play a crucial role in the adhesion of bacteria to the intestinal mucosa [48]. The auto-aggregation and co-aggregation abilities of the LAB isolates were evaluated to assess their potential for biofilm formation and pathogen exclusion. The auto-aggregation rates ranged from 32.72 ± 1.41% to 40.54 ± 6.95% at 4 h, 35.93 ± 1.62% to 66.31 ± 5.76% at 16 h, and 47.17 ± 1.72% to 84.29 ± 2.36% at 24 h of incubation (Figure 6A). Among the isolates, *L. brevis* NKFS9 exhibited notably high auto-aggregation percentages of 40.54 ± 6.95% and 66.31 ± 5.76% at 4 and 16 h, respectively. After 24 h of incubation, *L. brevis* NKFS8 displayed the highest auto-aggregation percentage (84.29 ± 2.36%), exceeding the reference strain *L. casei* ATCC 393, which achieved 77.56 ± 1.73%. Our findings do not align with the results reported by [55], who observed that *L. salivarius* ZJ614 exhibited the lowest autoaggregation ability of 60%. In this study, the auto-aggregation percentages were observed to increase with prolonged incubation times, demonstrating a time-dependent enhancement of intercellular adhesion. Ref. [56] also reported an increasing autoaggregation rate from 14.57–36.40% after 2 h to 29.57–59.54% during the fifth hour.

In terms of co-aggregation, all LAB isolates demonstrated strong interactions with pathogenic bacteria, with co-aggregation percentages ranging from 52.56 ± 1.80% to 79.69 ± 3.49% after 4 h of incubation (Figure 6B). *P. pentosaceus* NKFS11 exhibited the highest co-aggregation percentage against *S. aureus*, whereas *L. brevis* NKFS7 showed the lowest co-aggregation ability against *P. aeruginosa*. Proteins involved in bacterial aggregation have been widely characterized, including Esp (260 kDa) in *E. faecalis*, which promotes adhesion and biofilm formation [57], and Bap (250 kDa) in *S. aureus*, similarly associated with biofilm development [58]. In *Lactobacillus gasseri*, aggregation has been attributed to a 2 kDa peptide [59]. The elevated auto-aggregation percentages observed in this study suggest the potential involvement of these adhesion-promoting proteins, which play a critical role in adherence to intestinal epithelial cells, a fundamental trait contributing to their probiotic efficacy.

#### 3.4.4. Cell Surface Hydrophobicity

The cell surface hydrophobicity (CSH) of the LAB strains was evaluated to assess their ability to adhere to hydrocarbons, using n-hexadecane as the non-polar solvent. The hydrophobicity percentages ranged from 29.7% to 73.15%, indicating significant variability (*p* < 0.05)) among the isolates (Figure 7). Ref. [9] also reported a CSH percentage ranging from 40.5 ± 12.02 to 61.5 ± 3.54% of LAB isolates from the small intestine of broiler chickens. All *L. brevis isolates* exhibited the highest adherence, with CSH percentages of 73.15%, 70.12%, and 64.50%, respectively (*p* > 0.05). These findings are in contrast with the findings reported by [60].

High levels of hydrophobicity indicate the adhesion capacity of LAB to epithelial cells, with the elevated percentages observed in the tested isolates underscoring their potential as promising candidates [61]. Conversely, *P. pentosaceus* NKFS10, *S. salivarius* NKFS6, and *P. pentosaceus* NKFS11 demonstrated the lowest adherence to n-hexadecane, with CSH percentages of 29.70%, 38.49%, and 52.10%, respectively. The modulation of cell adhesion properties by various factors, including culture medium composition, temperature, and pH, is well documented [62].

## 4. Conclusions

The results obtained in this study indicate that the small intestine of broiler breeders serves as a pool for the isolation of potential probiotic bacteria. Out of the 39 LAB strains isolated, only six (*S. salivarius* NKFS6, *L. brevis* NKFS7, *L. brevis* NKFS8, *L. brevis* NKFS9, *P. pentasaceus* NKFS10, and *P. pentisaceus* NKFS11) demonstrated desirable probiotic attributes, tolerance to gastrointestinal conditions, significant antimicrobial activity against poultry pathogens, and beneficial surface properties such as autoaggregation, co-aggregation, and hydrophobicity. The lack of hemolytic activity further supports their safety. The phenotypic evidence of antibiotic resistance observed among these isolates raises significant concerns regarding their safety and regulatory acceptability. The notable trends in antibiotic resistance identified during phenotypic screening, coupled with the absence of molecular detection of resistance genes and the determination of minimum inhibitory concentrations (MICs), further complicate the characterization of resistance. This lack of molecular analysis limits the comprehensive understanding of antibiotic resistance mechanisms and raises critical issues related to safety and regulatory approval. Although the isolates demonstrated promising probiotic attributes, it is imperative to rigorously assess their safety profile, with particular emphasis on antimicrobial resistance. Furthermore, comprehensive in vivo studies are warranted to evaluate their functional efficacy, impact on gut health, and long-term effects prior to considering their inclusion in animal feed formulations. These assessments should align with established international guidelines, such as those outlined by the European Food Safety Authority (EFSA) and the FAO/WHO, to ensure compliance with global safety and regulatory frameworks.

## Figures and Tables

**Figure 1 microorganisms-13-01231-f001:**
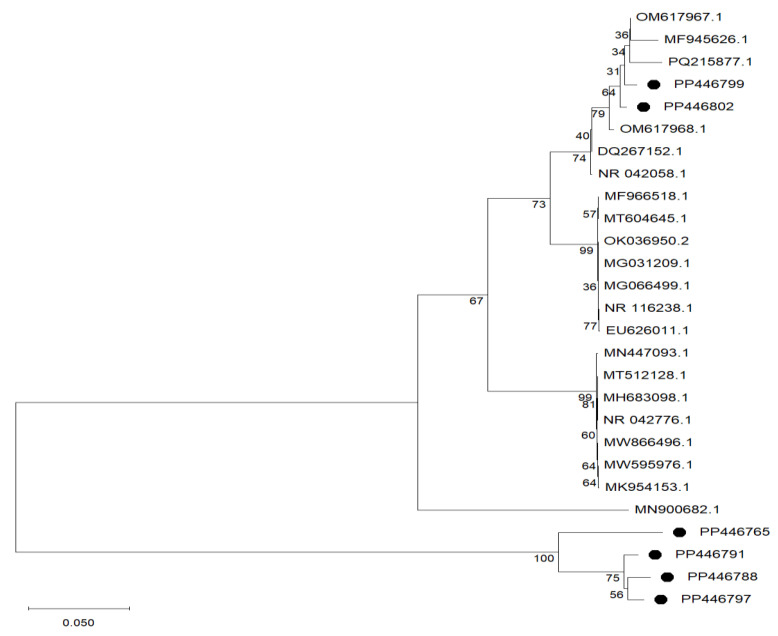
The phylogenetic tree of six LAB strains (in black dots) isolated from the small intestines of broiler breeders tree created by the neighbour-joining method using a bootstrap value of 1000.

**Figure 2 microorganisms-13-01231-f002:**
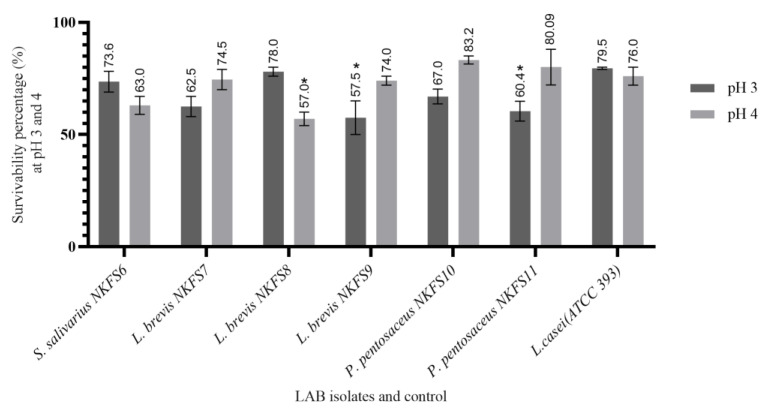
Survivability percentage of bacterial strains isolated from the small intestines of broiler breeders after 3 h exposure to pH 3 and pH 4. Reference strain: *Lactobacillus casei* (ATCC 393). *n* = 3 replicates. Values are means ± SEM. Significant differences between the means of LAB isolates and the reference strain are indicated as * *p* < 0.05.

**Figure 3 microorganisms-13-01231-f003:**
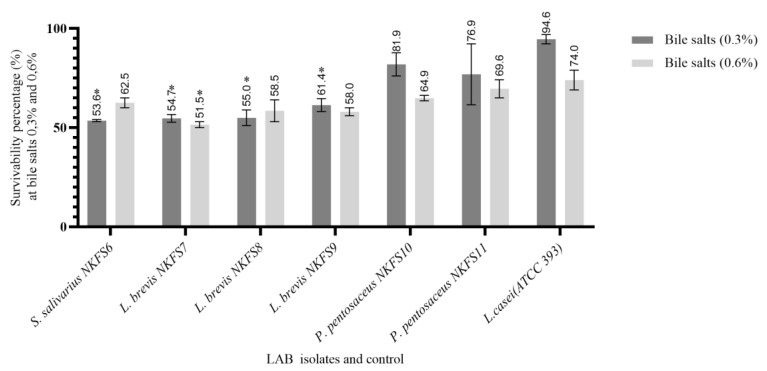
Survivability percentages of lactic acid bacterial isolates from the small intestines of broiler breeders after 3 h of exposure to bile salt concentrations of 0.3% (*w*/*v*) and 0.6% (*w*/*v*). Reference strain: *Lactobacillus casei* (ATCC 393). *n* = 3 replicates. Values are means ± SEM. Significant differences between the means of LAB isolates and the reference strain are indicated as * *p* < 0.05.

**Figure 4 microorganisms-13-01231-f004:**
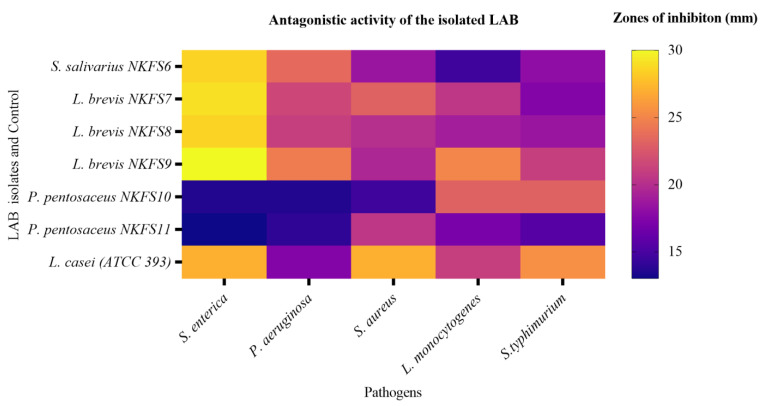
Heatmap showing the inhibition zones (in mm) produced by LAB isolates and a control (*L. casei* ATCC393) against *S. enterica*, *P. aeruginosa*, *S. aureus*, *L. monocyto* genes, and *S. typhimurium*. The intensity of the colour corresponds to the size of the inhibition zone, with yellow representing larger zones (>30 mm) and dark blue indicating minimal inhibition (<15 mm) with various poultry pathogens. Values are the mean of three replicates (*n* = 3).

**Figure 5 microorganisms-13-01231-f005:**
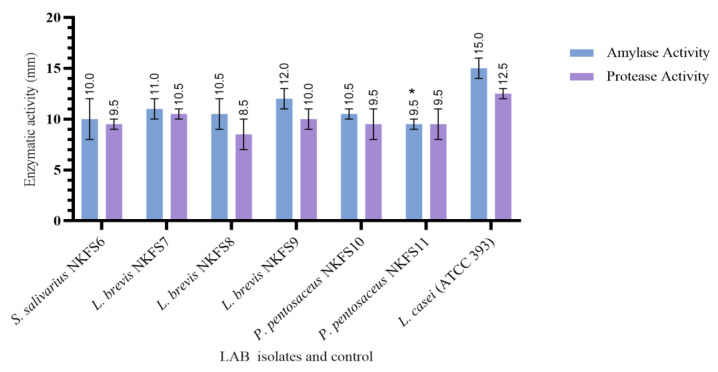
Extracellular enzymatic activity of LAB strains from the small intestine. Values are expressed as mean ± SEM. Significant differences between the means of LAB isolates and the reference strain are indicated as * *p* < 0.05.

**Figure 6 microorganisms-13-01231-f006:**
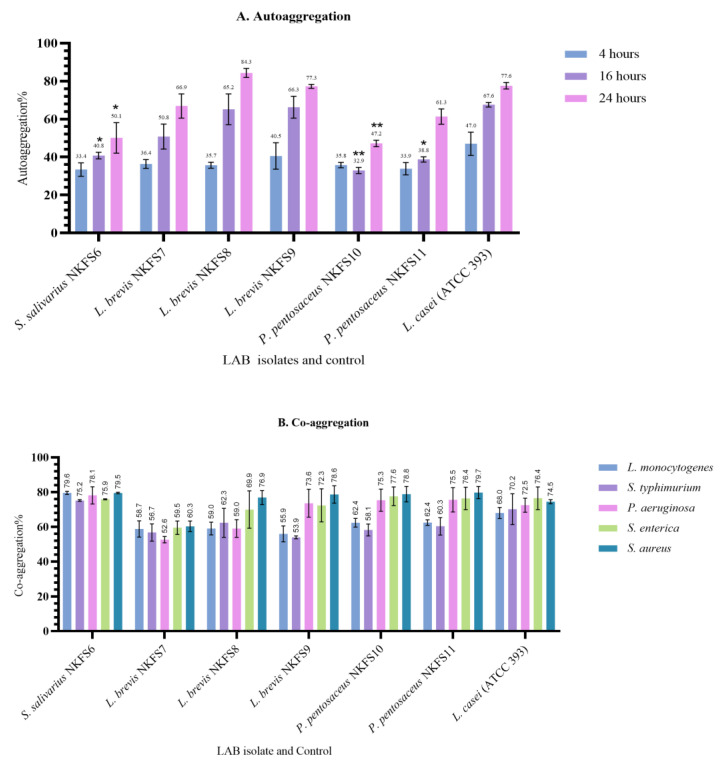
(**A**) Auto aggregation (%) of six LAB strains after 4, 16, and 24 h of incubation. (**B**) Co-aggregation (%) of LAB strains against various pathogens. Values are expressed as mean ± SEM. Significant differences between the means of LAB isolates and the reference strain are indicated as * *p* < 0.05 and ** *p* < 0.001.

**Figure 7 microorganisms-13-01231-f007:**
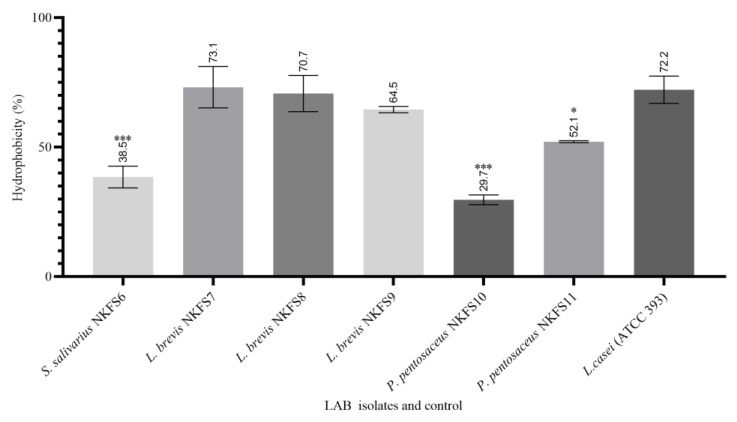
Cell surface hydrophobicity percentage (BATS%) of LAB isolates from the small intestines of broiler breeders to n-hexadecane. Reference strain: *Lactobacillus casei* (ATCC 393). *n* = 3 replicates. Values are means ± SEM. Significant differences between the means of LAB isolates and the reference strain are indicated as * *p* < 0.05 and *** *p* < 0.0001.

**Table 1 microorganisms-13-01231-t001:** Phenotypic characterization and 16S rDNA sequencing identification of LAB from the small intestine of broiler breeders.

Isolate ID	Morphology	Gram-Stain	Catalase Test	Phylogenetic Affiliation	GenBank Accession No.	Similarity
SI4	Cocci	+	−	*Streptococcus salivarius* NR042776.1	PP446765	100%
SI6	Bacilli	+	−	*Levilactobacillus brevis* NR116238.1	PP446788	99.85%
SI8	Bacilli	+	−	*Levilactobacillus brevis* NR116238.1	PP446791	99.85%
SI9	Bacilli	+	−	*Levilactobacillus brevis* NR116238.1	PP446797	99.85%
SI23	Cocci	+	−	*Pediococcus pentosaceus* NR042058.1	PP446799	99.58%
SI38	Cocci	+	−	*Pediococcus pentosaceus* NR042058.1	PP446802	99.51%

**Table 2 microorganisms-13-01231-t002:** Haemolytic activity and antibiotic susceptibility of distinct lactic acid bacteria isolated from the small intestine of broiler breeders to a range of antibiotics.

Isolate ID	Haemolytic Activity	GM	S	C	CRO	CIP	VA	E	AMP	P	OX	TE	NV
*S. salivarius* NKFS6	-	I	I	S	S	R	R	S	S	I	R	S	R
*L. brevis* NKFS7	-	R	R	S	S	R	R	S	S	I	R	S	R
*L. brevis* NKFS8	-	S	S	S	S	R	R	S	S	I	R	S	R
*L. brevis* NKFS9	-	R	I	S	S	R	R	S	S	I	R	S	R
*P. pentosaceus* NKFS10	-	R	R	R	R	R	R	R	R	R	R	R	R
*P. pentosaceus* NKFS11	-	R	R	S	S	R	R	S	S	R	I	S	R
*L. casei* (ATCC 393)	-	R	R	S	R	R	R	S	S	S	R	R	R

## Data Availability

The original contributions presented in this study are included in the article/Appendix A. Further inquiries can be directed to the corresponding authors.

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
