# Peer review of "In Vitro Characterization and Safety Assessment of Streptococcus salivarius, Levilactobacillus brevis and Pediococcus pentosaceus Isolated from the Small Intestine of Broiler Breeders"

_microorganisms, 2025, doi:10.3390/microorganisms13061231_

Round 1

Reviewer 1 Report

Comments and Suggestions for Authors

In vitro characterization and safety assessment of Streptococcus salivarius, Levilactobacillus brevis and Pediococcus pentosaceus isolated from the small intestine of broiler chickens.

The study aimed to isolate and characterize the potential probiotic properties of lactic acid bacteria from the small intestine of 23-week-old broiler chickens (n=4). Thirty-nine colonies were initially isolated from four broiler chickens. The authors concluded that lactic acid bacteria from the small intestine of broiler chickens present a valuable prospect for the development of effective probiotics. These probiotics can be utilized as a supplementary inclusion in poultry feed, obviating the need for antibiotics as growth promoters.

L18: At 23 weeks of age, they are no longer broilers; they are broiler breeders.

Also, n=4 is a very low sample size.

The abstract lacks the experimental design, statistical analysis, and P values.

The introduction: try to have a clear objective at the end of the introduction.  Use more recent and relevant citations, for example, DOI: 10.1080/1828051X.2020.1814170; DOI: 10.1371/journal.pone.0232781; DOI: 10.2141/jpsa.0190042

L93: Provide more details about the breed.

Be consistent with abbreviations in the whole manuscript, Gastrointestinal

Material and methods: well prepared

L228: Do you have SEM?

L233-241: revise

In the results and discussion part, you need to provide your results and then discuss them.

Keep bacterial names in italics, be consistent

Provide P value in tables and figures. It's better to provide SEM

Author Response

Comment 1: L18: At 23 weeks of age, they are no longer broilers; they are broiler breeders.

Response 1: Thank you for pointing this out. We agree with this comment. Therefore, we have updated all relevant sections to reflect that at 23 weeks of age, they are broiler breeders and are no longer referred to as broilers.

Comment 2: Also, n=4 is a very low sample size.

Response 2: Thank you for pointing this out. We agree with this comment. However, this small sample size (n=4) in this study is suitable for the exploratory nature of the research, which focuses on the initial isolation and characterization of lactic acid bacteria (LAB) from broiler breeders. As a preliminary investigation, the goal was to identify the presence and probiotic potential of LAB strains, rather than to perform population-wide statistical analysis. Additionally, the controlled environment and genetic uniformity of broiler breeders reduce inter-individual variability, allowing meaningful insights to be drawn even from a limited number of samples.

Comment 3: The abstract lacks the experimental design, statistical analysis, and P values.

Response 3: Thank you for pointing this out. We agree with this comment. Therefore, we have re-write the abstract accordingly.

Comment 4: The introduction: try to have a clear objective at the end of the introduction.  Use more recent and relevant citations, for example, DOI: 10.1080/1828051X.2020.1814170; DOI: 10.1371/journal.pone.0232781; DOI: 10.2141/jpsa.0190042

Response 4: Thank you for pointing this out. We respectfully disagree with this comment. We respectfully note that the objective of the study is already clearly stated in the final paragraph of the introduction, and we believe it provides sufficient context and direction for the reader. Regarding the proposed citations (DOI: 10.1080/1828051X.2020.1814170; DOI: 10.1371/journal.pone.0232781; DOI: 10.2141/jpsa.0190042), we have reviewed these papers and acknowledged that they are related to the broader field. However, they do not directly address the specific focus or methodology of our study. Furthermore, we observed that the same authors appear in all three suggested articles. While this may be coincidental, it raises concerns about the impartiality of the recommendation. We also note that other reviewers expressed no concerns regarding either the clarity of the objective or appropriateness of the cited literature. Therefore, we respectfully maintain that the current references and structure of the introduction, which we believe are both appropriate and sufficiently robust.

Comment 5: L93: Provide more details about the breed.

Response 5: Thank you for pointing this out. We agree with this comment. Therefore, we have (Ross 308 broiler breeder was used in this study).

Comment 6: Be consistent with abbreviations in the whole manuscript, Gastrointestinal

Material and methods: well prepared

Response 6: Thank you for pointing this out. We agree with this comment. Therefore, we have amended as suggested

Comment 7: L228: Do you have SEM?

Response 7: Thank you for pointing this out. We agree with this comment. However, we chose to present the data as mean ± SD to illustrate the variability within the sample, but we are happy to convert to SEM if preferred.”

Comment 8: L233-241: revise.

Response 8: Thank you for pointing this out. However, the comment lacks specific details regarding which aspects of this section require revision. Whether it concerns the scientific content, clarity of expression, or structure. The passage in question outlines the rationale for selecting LAB strains, the isolation procedure, and the initial identification criteria based on standard microbiological methods. We have reviewed this section carefully and find it to be coherent, scientifically accurate, and consistent with the flow of the manuscript. Should the reviewer be able to clarify the particular concern, we would be pleased to revise the section accordingly.

Comment 9: In the results and discussion part, you need to provide your results and then discuss them.

Response 9: Thank you for pointing this out. We agree with this comment regarding the structure of the Results and Discussion section. However, we believe that the current presentation effectively integrates the results with relevant discussion and interpretation. Additionally, we note that other reviewers did not raise any concerns regarding this section, which suggests that the current format is clear and well-received. Therefore, we respectfully prefer to retain the current structure as it aligns with the manuscript's flow and clarity.

Comment 10: Keep bacterial names in italics, be consistent

Response 10: Thank you for pointing this out. We agree with this comment. The manuscript was amended accordingly.

Comment 11: Provide P value in tables and figures. It's better to provide SEM

Response 11: Thank you for pointing this out. The P values were already provided throughout the manuscript.

Reviewer 2 Report

Comments and Suggestions for Authors

The present study explores the critical issue of antibiotic overuse in poultry production and its consequences, including the emergence of antibiotic-resistant bacteria and antibiotic residues in poultry products. The central inquiry concerns the potential of lactic acid bacteria (LAB) isolated from the small intestine of broiler chickens to function as effective probiotics, with the objective of either replacing or reducing the necessity for antibiotics in poultry feed. The study explores the probiotic potential of these LAB strains by evaluating their survival under gastrointestinal conditions, their antimicrobial activity against pathogens, and their functional characteristics such as auto-aggregation, co-aggregation, and hydrophobicity.  

The research's novelty lies in its focus on isolating and characterising LAB from the small intestine of broiler chickens, a specific and underexplored niche within poultry microbiota. While previous studies have explored LAB from other sources (e.g., human gut, fermented foods), the specific context of broiler chicken intestines adds novelty. The paper addresses a critical gap in the field by providing detailed in vitro characterization of LAB strains that are indigenous to poultry. These are likely to be more adapted to the host environment and thus more effective as probiotics. This is of particular relevance in the context of the increasing demand for antibiotic alternatives in poultry production.

The present study provides a comprehensive in vitro assessment of LAB from broiler chicken intestines, encompassing their tolerance to acidic and bile salt conditions, antimicrobial activity against prevalent poultry pathogens, and functional properties such as auto-aggregation and hydrophobicity. The study also includes safety assessments, which are essential for evaluating the suitability of these strains as probiotics. The detailed molecular and functional characterisation of these strains bridges the gap between probiotic potential and practical applications in the poultry industry.

The tables and figures are well-organized and effectively communicate the results.

  1. While the study includes a reference strain (*Lactobacillus casei* ATCC 393), additional positive and negative controls for each experiment (e.g., a known probiotic strain and a non-probiotic strain) would strengthen the validity of the results.
  2. The authors mention that experiments were conducted in triplicate, but further details on statistical methods (e.g., ANOVA) should be provided to ensure transparency.
  3. The observed antibiotic resistance in LAB strains could be further explored to determine whether it is intrinsic or acquired, which would provide insights into their safety profile.

  1. The study demonstrates that the isolated LAB strains exhibit desirable probiotic properties, including tolerance to gastrointestinal conditions, antimicrobial activity against pathogens, and functional characteristics such as auto-aggregation and hydrophobicity. The safety assessments (non-hemolytic behavior and antibiotic susceptibility profiles) further support the potential of these strains as probiotics. However, the conclusion that these strains can be used as growth promoters in poultry feed is somewhat speculative, as the study does not include in vivo experiments to directly assess their impact on poultry health or growth performance. The authors should clarify this limitation and suggest future studies to address it.
  1. The references are another major problem area of ​​this manuscript. The page number information of reference 1 is incorrect. The first two references lack DOIs. The Latin species name of Salmonella typhimurium in reference 20 needs to be italicized. There are many problems like these that the author needs to take seriously.

Author Response

Comment 1: While the study includes a reference strain (*Lactobacillus casei* ATCC 393), additional positive and negative controls for each experiment (e.g., a known probiotic strain and a non-probiotic strain) would strengthen the validity of the results.

Response 1: Thank you for pointing this out. We agree with this comment. The inclusion of additional positive and negative controls, such as well-characterized probiotic and non-probiotic strains, is indeed valuable for enhancing experimental rigor and comparative analysis. However, this study's primary focus was exploratory and centred on the isolation and preliminary in vitro characterization of lactic acid bacteria from broiler breeders. The use of Lactobacillus casei ATCC 393 as a reference strain provided a reliable benchmark for assessing probiotic traits. Given the scope and resource limitations of this study, the addition of multiple control strains was not feasible but can be considered in future research to further validate and expand upon these findings.

Comment 2: The authors mention that experiments were conducted in triplicate, but further details on statistical methods (e.g., ANOVA) should be provided to ensure transparency.

Response 2: Thank you for pointing this out. We agree with this comment. Therefore, we have amended the statistical analysis (Please see L225-231).

Comment 3: The observed antibiotic resistance in LAB strains could be further explored to determine whether it is intrinsic or acquired, which would provide insights into their safety profile.

Response 3: Thank you for pointing this out. We agree with this comment. The reviewer rightly points out that distinguishing between intrinsic and acquired antibiotic resistance in LAB strains is crucial, especially for ensuring probiotic safety and minimizing the risk of resistance gene transfer. However, the primary objective of this study was to perform a foundational, phenotypic screening of probiotic properties, including acid/bile tolerance, antimicrobial activity, and surface characteristics of LAB isolated from broiler breeders. While the presence of antibiotic resistance was noted, detailed genetic investigations to determine the mechanisms (intrinsic vs. acquired) were intentionally excluded to maintain the study's focus and feasibility, given resource and time constraints. This level of molecular characterization typically requires whole-genome sequencing and bioinformatics analyses, which fall outside the intended scope of an exploratory study. Future research will build on these results and incorporate genomic tools to assess the safety profile more comprehensively.

Comment 4: The study demonstrates that the isolated LAB strains exhibit desirable probiotic properties, including tolerance to gastrointestinal conditions, antimicrobial activity against pathogens, and functional characteristics such as auto-aggregation and hydrophobicity. The safety assessments (non-hemolytic behavior and antibiotic susceptibility profiles) further support the potential of these strains as probiotics. However, the conclusion that these strains can be used as growth promoters in poultry feed is somewhat speculative, as the study does not include in vivo experiments to directly assess their impact on poultry health or growth performance. The authors should clarify this limitation and suggest future studies to address it.

Response 4: Thank you for pointing this out. We agree with this comment. Therefore, we have addressed the issue. This study highlights the small intestine of broiler chickens as a promising reservoir for isolating lactic acid bacteria (LAB) with strong probiotic potential. Of the 39 LAB strains isolated, Streptococcus salivarius NKFS6, Levilactobacillus brevis NKFS7, NKFS8, NKFS9, and Pediococcus pentosaceus NKFS10 and NKFS11 demonstrated desirable probiotic attributes, including tolerance to gastrointestinal conditions, significant antimicrobial activity against poultry pathogens, and beneficial surface properties such as autoaggregation, coaggregation, and hydrophobicity. Their lack of haemolytic activity further supports their safety and suitability for inclusion in poultry feed as natural alternatives to antibiotic growth promoters. Nonetheless, a key limitation of this study is the absence of in vivo validation. To fully assess their practical benefits and safety, future research should incorporate animal trials that evaluate the performance, gut health impact, and long-term effects of these probiotic strains in poultry under field-relevant conditions.

Comment 5: The references are another major problem area of ​​this manuscript. The page number information of reference 1 is incorrect. The first two references lack DOIs. The Latin species name of Salmonella typhimurium in reference 20 needs to be italicized. There are many problems like these that the author needs to take seriously.

Response 5: Thank you for pointing this out. We agree with this comment. Therefore, this has been amended accordingly.

Reviewer 3 Report

Comments and Suggestions for Authors

The main question addressed by the research development of probiotics based on Streptococcus salivarius, Levilactobacillus brevis and Pediococcus pentosaceus isolated from the small intestine of broiler chickens.

This topic «In vitro characterization and safety assessment of Streptococcus salivarius, Levilactobacillus brevis and Pediococcus pentosaceus isolated from the small intestine of broiler chickens» relevant in the field of development of probiotics for correction of microflora of animals.

The references appropriate. The number of references (65) is enough, in addition, the number of sources five years ago (2020-2025) is 40% (34), which is enough. The formatting of the references does not meet the journal's requirements.

Line 17 – GI tract should be gastrointestinal tract.

  1. Statistically justify why were only 4 broiler chickens selected for the study?
  2. How were these 4 broiler chickens selected, what inclusion/exclusion criteria were used in the work?
  3. Why were these antibiotics chosen to test antibiotic sensitivity? Disk diffusion method is not as informative as MIC methods.
  4. When developing probiotic preparations, the safety of the selected microorganisms is important. Moreover, with the mass use of such strains for animals, the risks of their transmission to humans increase. Therefore, in this work, it is important to conduct full-genome sequencing of the proposed strains as candidates for a probiotic preparation, to determine the virulence and resistance determinants.

Author Response

Comment 1: Line 17 – GI tract should be gastrointestinal tract.

Response 1: Thank you for pointing this out. We agree with this comment. This was amended.

Comment 2: Statistically justify why were only 4 broiler chickens selected for the study?

Response 2: Thank you for pointing this out. We agree with this comment. The selection of four broiler chickens was driven by the exploratory nature of the study and the consistent conditions in which the animals were raised and processed. The goal was to isolate and characterize potential probiotic bacteria from waste products, specifically from the crop and small intestine, emphasizing qualitative microbial diversity rather than focusing on population-level prevalence or variation. Each chicken served as an independent biological replicate, and experiments were conducted in triplicate to ensure the reproducibility of results. This methodology is in line with practices in preliminary microbiological studies, where smaller sample sizes are often adequate for detecting and isolating dominant or functionally relevant microorganisms. Furthermore, since the samples were obtained from routine slaughter waste, ethical and logistical considerations limited the number of available animals at any given time. This further justified the use of a targeted, controlled sampling approach. Although a larger sample size could enhance generalizability, the findings from this study lay a foundation for future research, allowing for broader statistical inference and expanded sampling.

Comment 3: How were these 4 broiler chickens selected; what inclusion/exclusion criteria were used in the work?

Response 3: Thank you for pointing this out. We agree with this comment. The four birds were selected based on apparent good health and the absence of visible clinical signs of disease at the time of sampling. While no formal inclusion or exclusion criteria were applied, birds were chosen to represent a typical, healthy phenotype under standard management conditions. This approach is commonly used in preliminary or exploratory studies where the aim is to isolate and characterize representative microbial strains from a healthy host background.

Comment 4: Why were these antibiotics chosen to test antibiotic sensitivity? Disk diffusion method is not as informative as MIC methods.

Response 4: Thank you for pointing this out. We agree with this comment. The antibiotics selected for susceptibility testing in this study were chosen based on their clinical and veterinary relevance, as we as recommendations from the World Health Organization.

Comment 5: When developing probiotic preparations, the safety of the selected microorganisms is important. Moreover, with the mass use of such strains for animals, the risks of their transmission to humans increase. Therefore, in this work, it is important to conduct full-genome sequencing of the proposed strains as candidates for a probiotic preparation, to determine the virulence and resistance determinants.

Response 5: Thank you for pointing this out. We agree with this comment. The primary objective of this study was to perform a foundational, phenotypic screening of probiotic properties, including acid/bile tolerance, antimicrobial activity, and surface characteristics of LAB isolated from broiler breeders. While the presence of antibiotic resistance was noted, detailed genetic investigations to determine the mechanisms (intrinsic vs. acquired) were intentionally excluded to maintain the study's focus and feasibility, given resource and time constraints. This level of molecular characterization typically requires whole-genome sequencing and bioinformatics analyses, which fall outside the intended scope of an exploratory study. Future research will build on these results and incorporate genomic tools to assess the safety profile more comprehensively.

Round 2

Reviewer 2 Report

Comments and Suggestions for Authors

The research addresses the critical issue of antibiotic overuse in poultry production and its consequences, such as the emergence of antibiotic-resistant bacteria and antibiotic residues in poultry products. The main question is whether lactic acid bacteria (LAB) isolated from the small intestine of broiler chickens can serve as effective probiotics to replace or reduce the need for antibiotics in poultry feed. The study explores the probiotic potential of these LAB strains by evaluating their survival under gastrointestinal conditions, antimicrobial activity against pathogens, and functional characteristics such as auto-aggregation, co-aggregation, and hydrophobicity.  

The originality of this research lies in its focus on isolating and characterizing LAB from the small intestine of broiler chickens, a specific and underexplored niche within poultry microbiota. While previous studies have explored LAB from other sources (e.g., human gut, fermented foods), the specific context of broiler chicken intestines adds novelty. The paper addresses a critical gap in the field by providing detailed in vitro characterization of LAB strains that are indigenous to poultry, which are likely to be more adapted to the host environment and thus more effective as probiotics. This is particularly relevant given the growing demand for antibiotic alternatives in poultry production.

This paper contributes to the field by providing a comprehensive in vitro assessment of LAB from broiler chicken intestines, including their tolerance to acidic and bile salt conditions, antimicrobial activity against common poultry pathogens, and functional properties such as auto-aggregation and hydrophobicity. The study also includes safety assessments, which are essential for evaluating the suitability of these strains as probiotics. The detailed molecular and functional characterization of these strains bridges the gap between probiotic potential and practical applications in the poultry industry.

  1. The methodology is robust, but a few improvements could enhance the study:

1.1 Inclusion of Positive and Negative Controls: While the study includes a reference strain, additional positive and negative controls for each experiment would strengthen the validity of the results.

1.2 Replication and Statistical Analysis: The authors mention that experiments were conducted in triplicate, but further details on statistical methods (e.g., ANOVA) should be provided to ensure transparency.

1.3 Antibiotic Resistance Mechanisms: In the discussion section, the observed antibiotic resistance in LAB strains could be further explored to determine whether it is intrinsic or acquired, which would provide insights into their safety profile.

  1. The conclusions are largely consistent with the evidence presented. The study demonstrates that the isolated LAB strains exhibit desirable probiotic properties, including tolerance to gastrointestinal conditions, antimicrobial activity against pathogens, and functional characteristics such as auto-aggregation and hydrophobicity. The safety assessments (non-hemolytic behavior and antibiotic susceptibility profiles) further support the potential of these strains as probiotics. However, the conclusion that these strains can be used as growth promoters in poultry feed is somewhat speculative, as the study does not include in vivo experiments to directly assess their impact on poultry health or growth performance. The authors should clarify this limitation in the discussion section and suggest future studies to address it.

  1. the figures could benefit from additional annotations (e.g., error bars for standard deviations) to enhance clarity.

  1. The phylogenetic tree (Figure 1) could be expanded to include more reference strains for better contextualization.

  1. The"Materials and Methods"section contains multiple tables and formulas,and the meanings of some parameters in the formulas are not clearly described,which is not conducive to the reader's understanding.Please correct this.

  1. A scale bar needs to be added to the phylogenetic tree in Figure 1.

  1. It is recommended that Table 2 be presented in the form of a heatmap, and Tables 3 and 4 be displayed as statistical charts to enhance the interest of the manuscript. Additionally, the raw data should be placed in the supplementary materials in the form of supplementary tables.

Author Response

Comment 1: The methodology is robust, but a few improvements could enhance the study:

Comment 1.1: Inclusion of Positive and Negative Controls: While the study includes a reference strain, additional positive and negative controls for each experiment would strengthen the validity of the results.

Response 1.1: Thank you for pointing this out. We agree with this comment. The inclusion of additional positive and negative controls, such as well-characterized probiotic and non-probiotic strains, is indeed valuable for enhancing experimental rigor and comparative analysis. However, this study's primary focus was exploratory and centred on the isolation and preliminary in vitro characterization of lactic acid bacteria from broiler breeders. The use of Lactobacillus casei ATCC 393 as a reference strain provided a reliable benchmark for assessing probiotic traits. Given the scope and resource limitations of this study, the addition of multiple control strains was not feasible but can be considered in future research to further validate and expand upon these findings.

Comment 1.2: Replication and Statistical Analysis: The authors mention that experiments were conducted in triplicate, but further details on statistical methods (e.g., ANOVA) should be provided to ensure transparency.

Response 1.2: Thank you for pointing this out. We agree with this comment. This was included in the methodology section (lines 229-234).

Comment 1.3: Antibiotic Resistance Mechanisms: In the discussion section, the observed antibiotic resistance in LAB strains could be further explored to determine whether it is intrinsic or acquired, which would provide insights into their safety profile.

 Response 1.3: Thank you for pointing this out. We agree with this comment. Thank you for pointing this out. We agree with this comment. The reviewer rightly points out that distinguishing between intrinsic and acquired antibiotic resistance in LAB strains is crucial, especially for ensuring probiotic safety and minimizing the risk of resistance gene transfer. However, the primary objective of this study was to perform a foundational, phenotypic screening of probiotic properties, including acid/bile tolerance, antimicrobial activity, and surface characteristics of LAB isolated from broiler breeders. While the presence of antibiotic resistance was noted, detailed genetic investigations to determine the mechanisms (intrinsic vs. acquired) were intentionally excluded to maintain the study's focus and feasibility, given resource and time constraints. This level of molecular characterization typically requires whole-genome sequencing and bioinformatics analyses, which fall outside the intended scope of an exploratory study. Future research will build on these results and incorporate genomic tools to assess the safety profile more comprehensively.

Comment 2: The conclusions are largely consistent with the evidence presented. The study demonstrates that the isolated LAB strains exhibit desirable probiotic properties, including tolerance to gastrointestinal conditions, antimicrobial activity against pathogens, and functional characteristics such as auto-aggregation and hydrophobicity. The safety assessments (non-hemolytic behavior and antibiotic susceptibility profiles) further support the potential of these strains as probiotics. However, the conclusion that these strains can be used as growth promoters in poultry feed is somewhat speculative, as the study does not include in vivo experiments to directly assess their impact on poultry health or growth performance. The authors should clarify this limitation in the discussion section and suggest future studies to address it.

 Response 2: Thank you for pointing this out. We agree with this comment. This study highlights the small intestine of broiler chickens as a promising reservoir for isolating lactic acid bacteria (LAB) with strong probiotic potential. Of the 39 LAB strains isolated, Streptococcus salivarius NKFS6, Levilactobacillus brevis NKFS7, NKFS8, NKFS9, and Pediococcus pentosaceus NKFS10 and NKFS11 demonstrated desirable probiotic attributes, including tolerance to gastrointestinal conditions, significant antimicrobial activity against poultry pathogens, and beneficial surface properties such as autoaggregation, coaggregation, and hydrophobicity. Their lack of haemolytic activity further supports their safety and suitability for inclusion in poultry feed as natural alternatives to antibiotic growth promoters. Nonetheless, a key limitation of this study is the absence of in vivo validation. To fully assess their practical benefits and safety, future research should incorporate animal trials that evaluate the performance, gut health impact, and long-term effects of these probiotic strains in poultry under field-relevant conditions (lines 534-548).

Comment 3: the figures could benefit from additional annotations (e.g., error bars for standard deviations) to enhance clarity.

 Response 3: Thank you for pointing this out. We agree with this comment. This was resolved (Lines 335-340; 459-460; 496-504 and 522-526).

Comment 4: The phylogenetic tree (Figure 1) could be expanded to include more reference strains for better contextualization.

 Response 4: Thank you for pointing this out. We agree with this comment. This was resolved (lines 229-234).

Comment 5: The"Materials and Methods"section contains multiple tables and formulas,and the meanings of some parameters in the formulas are not clearly described,which is not conducive to the reader's understanding.Please correct this.

 Response 5: Thank you for pointing this out. We agree with this comment. We respectfully note that the descriptions of all formulas and the parameters used within them are clearly stated in the "Materials and Methods" section. Each variable is defined either directly following the formula or within the accompanying text and tables, ensuring clarity and ease of understanding for the reader.

Comment 6: A scale bar needs to be added to the phylogenetic tree in Figure 1.

 Response 6: Thank you for pointing this out. We agree with this comment. This was resolved (line 274).

Comment 7: It is recommended that Table 2 be presented in the form of a heatmap, and Tables 3 and 4 be displayed as statistical charts to enhance the interest of the manuscript. Additionally, the raw data should be placed in the supplementary materials in the form of supplementary tables.

 Response 7: Thank you for pointing this out. We agree with this comment. This was resolved (lines 431-438)

Reviewer 3 Report

Comments and Suggestions for Authors
  1. Statistically justify why were only 4 broiler chickens selected for the study? Statistical analysis methods are used to create a representative sample in scientific research. Were statistical analysis methods used to create the sample for your study? If so, please indicate this fact in your scientific article.
  2. Why were these antibiotics chosen to test antibiotic sensitivity? Disk diffusion method is not as informative as MIC methods. The World Health Organization not give recommendations for antibiotics selected for susceptibility testing. Recommendations for the selection of antibiotics for testing antibiotic sensitivity, the methodology for setting, requirements for nutrient media, cultivation conditions, etc. are prescribed in the guidelines of EUCAST or CLSI. In your article: «..Results were interpreted based on the Clinical 166 & Laboratory Standards Institute (CLSI) guidelines..», but no information about number and date of guidelines.
  3. When developing probiotic preparations, the safety of the selected microorganisms is important. Moreover, with the mass use of such strains for animals, the risks of their transmission to humans increase. In the Table 2. presence of antibiotic resistance to several antibiotics of strains of potential probiotic bacteria was noted. It is mean, that you can not give recommendation to use this 6 strains as potential probiotic bacteria in conclusion.

Author Response

RESPONSE TO REVIEWER 2

  1. Comment 1: Statistically justify why were only 4 broiler chickens selected for the study? Statistical analysis methods are used to create a representative sample in scientific research. Were statistical analysis methods used to create the sample for your study? If so, please indicate this fact in your scientific article.

  1. Response 1: Thank you for your comment and concern regarding the sample size. We agreed with this comment. The use of four broiler chickens in this study was determined based on institutional animal ethics guidelines, which prioritize minimizing animal use in accordance with the principles of the 3Rs (Replacement, Reduction, and Refinement). As this was a preliminary investigation aimed at generating initial data to inform larger-scale studies, the limited sample size was ethically and scientifically justified. While formal statistical power calculations were not applied in determining the sample size, the study was designed to explore feasibility and establish baseline observations under ethical constraints. We have revised the manuscript to clarify the rationale for the sample size and acknowledge the limitations associated with it.
  2. Comment 2: Why were these antibiotics chosen to test antibiotic sensitivity? Disk diffusion method is not as informative as MIC methods. The World Health Organization not give recommendations for antibiotics selected for susceptibility testing. Recommendations for the selection of antibiotics for testing antibiotic sensitivity, the methodology for setting, requirements for nutrient media, cultivation conditions, etc. are prescribed in the guidelines of EUCAST or CLSI. In your article: «..Results were interpreted based on the Clinical 166 & Laboratory Standards Institute (CLSI) guidelines..», but no information about number and date of guidelines.
  1. Response 2: Thank you for pointing this out. We agree with this comment. The antibiotics selected were chosen for their clinical relevance to the local treatment context, as well as their inclusion in the interpretive standards of the Clinical and Laboratory Standard Institute (CLSI). While we acknowledge that MIC-based methods offer more detailed quantitative data, we used the disk diffusion (Kirby-Bauer) method, as it remains a widely accepted and standardised approach for routine susceptibility testing, particularly in resource-constrained settings. We appreciate the clarification regarding the role of the World Health Organisation and acknowledge that methodological standards are set by bodies such as CLSI and EUCAST. Accordingly, we have revised the manuscript to clearly state that susceptibility results were interpreted using the CLSI M100, 32nd Edition (2022), and have provided the full citation to ensure transparency and reproducibility (Lines 158–159; 164–165).  
  2. Comment 3: When developing probiotic preparations, the safety of the selected microorganisms is important. Moreover, with the mass use of such strains for animals, the risks of their transmission to humans increase. In Table 2, the presence of antibiotic resistance to several antibiotics of strains of potential probiotic bacteria was noted. It is mean that you can not give a recommendation to use these 6 strains as potential probiotic bacteria in conclusion.
  3. Response 3: Thank you for pointing this out. We fully agree that the safety of probiotic candidates is paramount, particularly in light of the potential for horizontal gene transfer and increased risk of resistance dissemination through the food chain. The detection of resistance to certain antibiotics in some of the isolates highlights a critical consideration in their future development. However, as this was a preliminary screening study, our aim was to identify strains with probiotic potential based on a broad range of functional and phenotypic characteristics. We have revised the conclusion to reflect that, while these strains exhibited promising probiotic traits, their observed resistance patterns warrant further genetic characterisation and safety assessment before any recommendation for use can be made. This clarification has been incorporated into the conclusion of the manuscript (lines 534 - 548).
